# Multi-level interaction between HIF and AHR transcriptional pathways in kidney carcinoma

Véronique N Lafleur[1], Silvia Halim[1], Hani Choudhry[2], Peter J Ratcliffe[3,4] ⬥, David R Mole[1] ⬥

**Hypoxia-inducible factor (HIF) and aryl hydrocarbon receptor (AHR) are members of the bHLH-PAS family of transcription factors that underpin cellular responses to oxygen and to endogenous and exogenous ligands, respectively, and have central roles in the pathogenesis of renal cancer. Composed of heterodimers, they share a common HIF-1β/ARNT subunit and similar DNA-binding motifs, raising the possibility of crosstalk between the two transcriptional pathways. Here, we identify both general and locus-specific mechanisms of interaction between HIF and AHR that act both antagonistically and cooperatively. Specifically, we observe competition for the common HIF-1β/ARNT subunit, in cis synergy for chromatin binding, and overlap in their transcriptional targets. Recently, both HIF and AHR inhibitors have been developed for the treatment of solid tumours. However, inhibition of one pathway may promote the oncogenic effects of the other. Therefore, our work raises important questions as to whether combination therapy targeting both of these pro-tumourigenic pathways might show greater efficacy than targeting each system independently.**

## Introduction

The basic helix–loop–helix (bHLH) PER-ARNT-SIM (PAS) family of proteins are a closely related group of transcription factors that have important roles in both the physiological adaptation to environmental stimuli and in the pathogenesis of cancer (1, 2, 3). Although each has its own repertoire of transcriptional targets, their operation as heterodimers that use common dimerization partners within distinct subfamilies raises the possibility of complex interactions.

Notably, the hypoxia-inducible factor (HIF) responds to the hypoxic tumour microenvironment, which is a common feature of many solid cancers, in addition to being directly activated as a consequence of mutation in oncogenic and tumour suppressor pathways (4, 5). Dysregulation of HIF is linked to transcriptional

programs with key roles in tumourigenesis and has direct implications for patient prognosis (6, 7). The most direct and profound up-regulation of the HIF pathway occurs in clear cell renal cell carcinoma (ccRCC), the most prevalent form of kidney cancer (8). This is associated with bi-allelic inactivation of the von Hippel Lindau (VHL) tumour suppressor, which forms part of an E3 ubiquitin ligase complex that targets HIF-α subunits (HIF-1α and HIF-2α; also known as EPAS-1) for proteasomal degradation in the presence of oxygen (9). As a result of VHL loss, physiological HIF-α degradation is disrupted, leading to constitutive activation of HIF. After stabilisation, the HIF-α proteins interact with HIF-1β (also known as ARNT; aryl hydrocarbon receptor nuclear translocator) to form functional HIF-1 and HIF-2 heterodimers that bind to chromatin (10). Despite a common consensus DNA-binding motif, HIF-1 and HIF-2 manifest distinct and overlapping binding patterns and transcriptional outputs and have opposing functions in ccRCC biology (11, 12, 13). Indeed, although HIF-2 is pro-tumourigenic, with its targeting recently approved as a treatment for VHL syndrome-associated tumours (14), HIF-1 is generally considered a tumour suppressor in ccRCC (15). Consequently, the overall effect of HIF activation on ccRCC tumourigenesis results from fine-tuning this balance between the HIF-1 and HIF-2 transcriptional outputs (16, 17, 18, 19).

The aryl hydrocarbon receptor (AHR) is a ligand-activated bHLH-PAS protein that acts as a mediator of metabolic responses to environmental contaminants, dietary compounds, and endogenous metabolites (20). Many of these contaminants, including cigarette smoke, polycyclic aromatic hydrocarbons, and halogenated aromatic hydrocarbons, are known carcinogens, with AHR being required for polycyclic aromatic hydrocarbon– and halogenated aromatic hydrocarbon–induced toxicities (2, 21, 22, 23, 24, 25). Like HIF, AHR can also be modulated by the tumour microenvironment and has critical roles in cancer progression, tumour aggressiveness, patient prognosis, and in tumour immunity (26, 27). AHR also dimerizes with HIF-1β/ARNT, and current evidence suggests that HIF and AHR compete directly for binding to HIF-1β/ARNT (28). However, observations also point to a more complex and context-dependent crosstalk that includes gene-specific cooperative events (29).

[1]NDM Research Building, University of Oxford, Old Road Campus, Oxford, UK   [2]Department of Biochemistry, Faculty of Science, Center of Innovation in Personalized Medicine, King Fahd Center for Medical Research, King Abdulaziz University, Jeddah, Saudi Arabia   [3]Ludwig Institute for Cancer Research, University of Oxford, Old Road Campus, Oxford, UK   [4]The Francis Crick Institute, London, UK

Correspondence: david.mole@ndm.ox.ac.uk

As an excretory organ, the kidneys are highly exposed to carcinogens and AHR has been found to be up-regulated in ccRCC tumours, in which it is associated with pathological tumour stage, histological grade, and poor prognosis (30). This raises the possibility of a consequential interplay between the AHR and HIF pathways in ccRCC. However, it remains unclear how the context-dependent competition between HIF and AHR directly impacts their chromatin binding abilities genome wide and whether it can involve locus-specific regulation or cooperation. To better understand these events and their integration for pathway outputs in the context of ccRCC, we have used chromatin immunoprecipitation coupled to next-generation DNA sequencing (ChIP-seq) to examine pan-genomic patterns of HIF-1$\alpha$, HIF-2$\alpha$, AHR, and HIF-1$\beta$/ARNT binding under co-activating stimuli and have linked these findings to gene expression changes analysed by RNA-sequencing. Our work reveals direct competition between HIF-$\alpha$ and AHR proteins for interaction with HIF-1$\beta$/ARNT, but that the symmetry of this competition is sensitive to the relative stoichiometry of the transcription factors. In ccRCC cells, we observed bi-directional competition between HIF and AHR that leads to a widespread antagonism of chromatin binding and to global downstream changes in gene expression. Strikingly, this competition is diminished at regions bound by both HIF and AHR, where binding of one transcription factor tends to reinforce binding of the other, and at a small set of genes that are direct transcriptional targets of both transcription factors. Overall, this reveals an unexpected complexity in the overlap and interaction between the HIF and AHR transcriptional programs. Finally, we provide evidence of this antagonistic relationship in ccRCC tumours, which display an overall negative correlation between HIF- and AHR-target gene expression. These findings shed light on the complex and direct crosstalk between the HIF and AHR transcriptional pathways, which encompasses both a widespread inhibitory interaction and specific cooperative events in ccRCC. Given the important role of HIF and AHR as regulators of cancer progression, therapeutic inhibitors have now been developed to target each pathway (14, 26). Our work, therefore, has important therapeutic implications because inhibition of one pathway may promote the oncogenic effects of the other. This raises the possibility that targeting both pathways simultaneously may lead to enhanced synthetic lethality.

## Results

### Uni- and bi-directional competition between HIF-$\alpha$ and AHR in different cell lines

We first surveyed a panel of cell lines for crosstalk between the HIF and AHR transcriptional pathways, using the expression of canonical target genes for each transcription factor. To start, we determined optimal conditions for target gene induction using RCC4 renal cancer cells reconstituted with WT VHL (RCC4+VHL) to restore oxygen-dependent HIF-$\alpha$ regulation. Firstly, we found maximal or near maximal induction of HIF-target genes after incubation in 0.5% hypoxia for 6 h that was maintained at 18 h, when compared with normoxia (ambient oxygen) (Fig S1A). Secondly, to activate AHR, we used two endogenous AHR

ligands—ITE, an indole-based dietary ligand, and FICZ, a tryptophan photo-metabolite. Incubation of RCC4+VHL cells with 100 nM ITE resulted in a time course of AHR-target gene induction which was similar to that of HIF, whereas incubation of these cells with 10 nM FICZ resulted in a more transient AHR-target gene induction that was maximal at 6 h and waning by 18 h, when compared with DMSO (vehicle control) (Fig S1B). Therefore, we chose to examine the 6-h time point for these and subsequent experiments.

In four of the cell lines analysed for crosstalk—RCC4+VHL, HKC8, HepG2, and PC3—incubation in hypoxia resulted in inhibition of the canonical AHR-target gene *CYP1B1* (Fig 1A). No significant effect was seen in 786O+VHL cells, which express only the HIF-2$\alpha$ isoform of HIF. Conversely, hypoxic induction of the canonical HIF-target gene *NDRG1* was largely unaffected by treatment with ITE in HKC8, HepG2, and PC3 cells (Fig 1B). However, modest suppression of hypoxic *NDRG1* was observed in RCC4+VHL and 786O+VHL cells incubated with ITE. These cell lines have particularly high levels of AHR compared with HIF-$\alpha$, suggesting that the stoichiometry may favour bi-directional crosstalk (Fig S1C).

To explore this further, we examined additional HIF- and AHR-target genes, and other stimuli, in RCC4+VHL cells. In addition to *CYP1B1*, the ITE-mediated induction of *ALDH1A3*, *BMF*, and *ASB2* was suppressed in hypoxia (Fig 1C), whereas the induction of *CYP27A1* was unaffected. Similarly, FICZ-mediated induction of *CYP1B1*, *BMF*, and *ASB2* was also suppressed in hypoxia (Fig S1D). Furthermore, the HIF-prolyl hydroxylase inhibitor FG4592 induced HIF-1$\alpha$ and HIF-2$\alpha$ independently of hypoxia (Fig S1E) and resulted in down-regulation of all five AHR-target genes in ITE-treated cells and 4/5 genes in FICZ-treated cells (Fig S1F and G). This strongly implicates HIF rather than non-HIF pathways in the hypoxic suppression of AHR-target genes. However, to test this more formally, we examined the effect of hypoxia on the ITE-mediated induction of *CYP1B1* in HKC8 cells deficient in HIF-1$\alpha$, HIF-2$\alpha$, or both isoforms (Fig 1E). Effective knockout of these transcription factors was confirmed by abolition of hypoxic induction of HIF protein and HIF-target genes (Fig S1H–K). Cells lacking both HIF-1$\alpha$ and HIF-2$\alpha$ isoforms did not show any effect of hypoxia on ITE-mediated *CYP1B1* expression, confirming its dependence on HIF. Furthermore, in HKC8 cells, the hypoxic suppression of *CYP1B1* expression appeared largely mediated by HIF-1$\alpha$ because cells lacking this isoform exhibited no hypoxic regulation of *CYP1B1*, whereas loss of HIF-2$\alpha$ resulted in a more modest diminution in the effect of hypoxia.

Conversely, in addition to *NDRG1*, modest ITE-mediated suppression of the HIF-target genes *PFKFB4* and *INSIG2* was observed in hypoxia (Fig 1D), whereas FICZ led to down-regulation of *NDRG1* alone (Fig S1L). Furthermore, when HIF was stimulated with FG4592, neither ITE nor FICZ had a significant impact upon the induction of HIF-target genes (Fig S1M and N). Thus, although the crosstalk between HIF and AHR in RCC4+VHL cells operates in both directions, under the conditions used, HIF has a larger effect on AHR-target genes than AHR does on HIF-target genes.

### Direct competition between HIF-$\alpha$ and AHR for HIF-1$\beta$/ARNT dimerization in ccRCC cells

We next explored the mechanisms underlying the crosstalk between the HIF and AHR transcriptional pathways in RCC4+VHL cells.

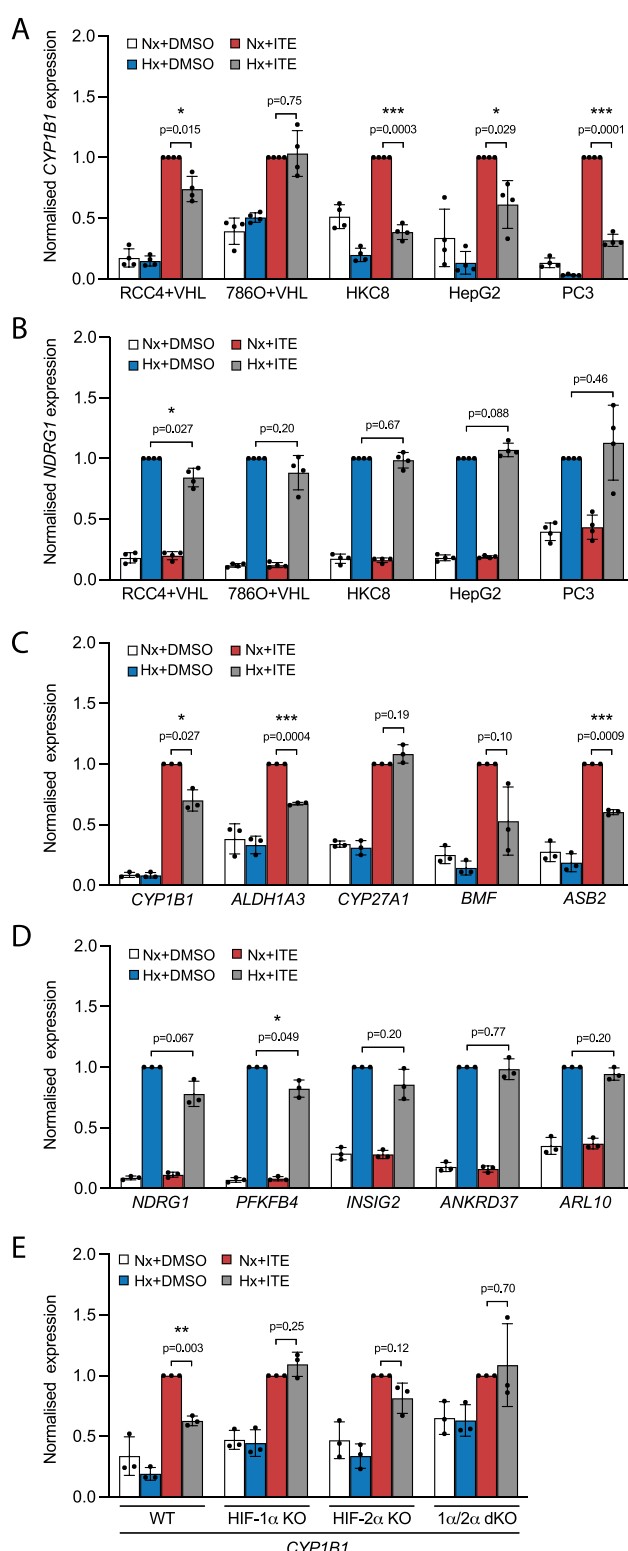

**Figure 1. Context-specific crosstalk between HIF and AHR.**
**(A, B)** RT-qPCR analysis of (A) *CYP1B1* and (B) *NDRG1* mRNA expression in RCC4+VHL, 786O+VHL, HKC8, HepG2, and PC3 cells subjected to incubation for 6 h in control conditions, 0.5% hypoxia, 100 nM ITE, or both. **(C, D)** RT-qPCR analysis of (C) AHR-target genes and (D) HIF-target genes in RCC4+VHL cells subjected to the same conditions as above. **(E)** RT-qPCR analysis of *CYP1B1* mRNA expression

The antagonistic relationship between HIF and AHR has previously been linked to competition for the common dimerization partner HIF-1β/ARNT, which may be limiting under specific circumstances (28). Therefore, we first assessed the competition between the HIF-α and AHR proteins for binding to HIF-1β/ARNT in RCC4+VHL cells. We initially determined optimal conditions for HIF-α and AHR induction in these cells. Firstly, exposure of RCC4+VHL cells to 6 h of 0.5% hypoxia resulted in maximal stabilisation of both HIF-1α and HIF-2α proteins (Fig S2A). Secondly, upon ligand binding, cytoplasmic AHR rapidly translocates to the nucleus and binds chromatin in complex with HIF-1β/ARNT. It subsequently undergoes protease-mediated degradation as part of a negative feedback mechanism (31). We therefore used translocation of AHR to the nucleus as a marker of its activation and observed rapid and transient increases in nuclear levels in response to 100 nM ITE and 10 nM FICZ that were maximal at 30 min (Fig S2B). Cells were therefore incubated for 6 h in either 0.5% hypoxia or normoxia, followed by treatment with ITE, FICZ, or DMSO (control) for the final 30 min (Fig 2A and B). These conditions resulted in strong induction of binding of each transcription factor to chromatin (Fig S2C and D).

Nuclear and cytoplasmic extracts were then prepared from these cells and immunoprecipitated using antibodies directed against HIF-1β/ARNT (Figs 2A and B and S3A). Although AHR, HIF-1α, and HIF-2α were all present in both the nucleus and the cytoplasm, the fraction bound to HIF-1β/ARNT was almost exclusively nuclear (Fig 2A and B). This is consistent with the previously described role for HIF-1β/ARNT as a nuclear translocator and a chromatin binding partner. Dimerization of HIF-1α and HIF-2α with HIF-1β/ARNT was observed under hypoxic incubation, whereas dimerization of AHR with HIF-1β/ARNT was strongly induced by both ITE and FICZ treatments (Fig 2A and B). Strikingly, ITE and FICZ treatments both reduced HIF-1α-HIF-1β/ARNT and HIF-2α-HIF-1β/ARNT dimer formation in hypoxic cells. Conversely, hypoxic incubation repressed AHR-HIF-1β/ARNT dimer formation in both ITE- and FICZ-treated cells even more strongly. Comparable inhibition of the AHR-HIF-1β/ARNT dimer was observed after stabilisation of HIF-α proteins by the HIF-prolyl hydroxylase inhibitor FG4592 (Fig S3B). Interestingly, relocation of AHR from nuclear to cytoplasmic cell fractions was observed under HIF-stabilising conditions, even in the presence of ITE and FICZ (Figs 2A and B and S3A and B). Thus, AHR nuclear localisation appears to correlate with its ability to interact with HIF-1β/ARNT and therefore to be repressed by competition with HIF-α. This suggests that the competition between HIF-α and AHR not only alters the ability of AHR to form an active transcription factor in the nucleus but also increases its cytoplasmic localisation, which may indirectly affect non-canonical AHR functions. Furthermore, re-oxygenation of cells after combined hypoxic and ITE treatments led to rapid degradation of the HIF-α proteins and to complete restoration of the AHR-HIF-1β/ARNT dimer (Fig S3C). Thus, the HIF-α and AHR proteins are able to compete reciprocally for HIF-1β/ARNT in RCC4+VHL cells, with HIF-α stabilisation having a greater impact

in control HKC8 cells and HKC8 cells deficient in HIF-1α, HIF-2α, or both. Data information: data are presented as mean ± SD of four (A, B) or three (C, D, E) biological replicates. \*P ≤ 0.05, \*\*P ≤ 0.01, \*\*\*P ≤ 0.001 (paired *t*-test).

A

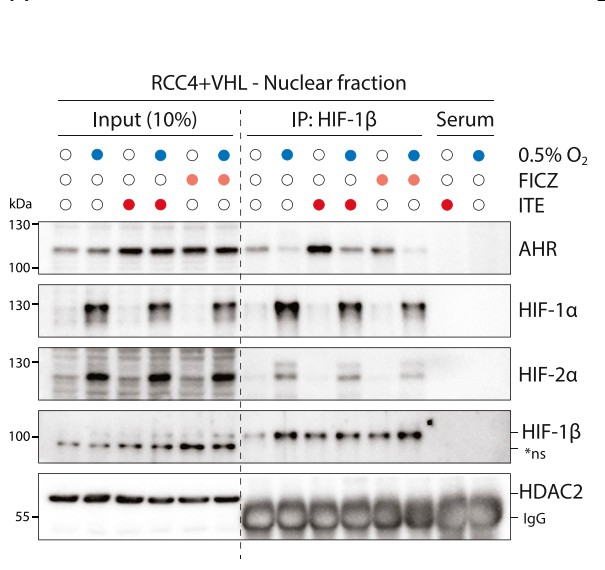

B

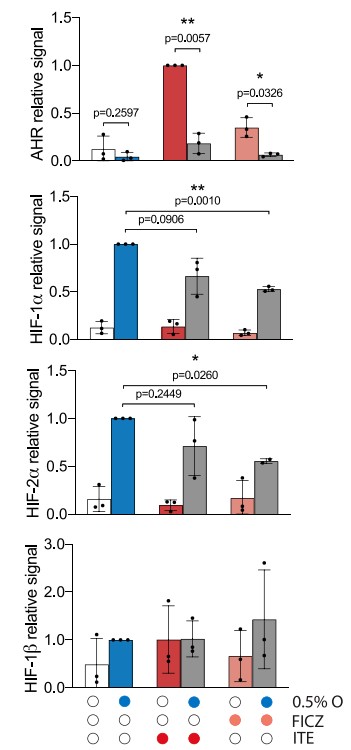

**Figure 2. HIF and AHR compete for HIF-1β/ARNT.**
RCC4+VHL cells were subjected to the indicated conditions and nuclear extracts were prepared and immunoprecipitated using antibodies directed against HIF-1β/ARNT (rabbit pAb NB100-110). **(A)** Input samples and immunoprecipitates were then immunoblotted for AHR, HIF-1α, HIF-2α, HIF-1β/ARNT (mouse mAb NB100-124), and HDAC2. **(B)** Quantitation of immunoprecipitation results from three biological replicates presented as mean ± SD *$P$ ≤ 0.05, **$P$ ≤ 0.01 (paired $t$-test). *ns, non-specific band.

on AHR binding than vice versa. Conversely, in HepG2 cells, in which crosstalk at the level of gene expression was found to be uni-directional, with HIF induction suppressing AHR-target gene expression but not vice versa, induction of HIF-α in hypoxia inhibited binding of AHR to HIF-1β/ARNT but induction of AHR did not inhibit HIF dimerization (Fig S3D). These cell type differences suggest that the bi-directionality or uni-directionality of the crosstalk between HIF and AHR is unlikely to be a consequence of inherent properties of HIF-α and AHR and more likely to result from differences in stoichiometry of the two transcription factors.

### Pan-genomic antagonism between HIF and AHR chromatin binding

Although competition between HIF-α and AHR for binding to HIF-1β/ARNT would likely result in widespread suppression of signaling in response to activation of the opposing factor, our analysis of target gene expression has shown heterogenous responses at individual target genes (Fig 1). Therefore, to further assess locus-specific crosstalk between HIF and AHR across the genome, we first performed ChIP-seq analysis of HIF-1α, HIF-2α, HIF-1β/ARNT, and AHR chromatin binding. Experiments were performed on RCC4+VHL cells incubated in either 0.5% hypoxia or normoxia for 6 h and treated with ITE, FICZ, or DMSO (control) for the final 30 min. All experiments were performed in duplicate and compared with input chromatin control samples. Only peaks identified by both the MACS and TPIC peak callers in both replicates of each condition were used for analysis.

Preliminary analysis was performed to examine the pairwise correlation between samples. The signal intensity at each peak (averaged between the two replicates) was plotted as a heat map

($r^2$ = 1, dark green, $r^2$ = 0, white), and samples were grouped using unsupervised hierarchical clustering (Fig S4). Three distinct clusters were identified, composed of samples from each of the HIF-1α, HIF-2α, and AHR ChIP-seq analyses. Samples from the HIF-1β/ARNT analyses clustered with the AHR analyses when cells were stimulated with ITE or FICZ alone but clustered with the HIF-1α analyses when cells were treated with hypoxia, even in the presence of ITE or FICZ. This indicates that HIF-1β/ARNT binding correlates more closely with HIF-1α binding than with AHR binding when both factors are present. Clustering of the hypoxic HIF-1β/ARNT analyses with HIF-1α rather than HIF-2α likely reflects the greater number of HIF-1 sites identified (n = 828) compared with HIF-2 sites (n = 401).

Next, canonical AHR binding sites (those with AHR and HIF-1β/ARNT signal in both replicates of a condition) were identified in normoxic cells treated with either DMSO, ITE or FICZ (Fig 3A). In total, 1,176 sites were identified in one or more condition. As expected, binding of AHR was strongly induced across these sites by both ITE and FICZ, with concomitant increases also seen for HIF-1β/ARNT signal (Fig S5A–K). Signal intensity for both AHR and HIF-1β/ARNT correlated very strongly between the two treatments ($r^2$ = 0.96 and 0.94, respectively) with all sites having comparable signal with either ITE or FICZ (Fig S5L and M). However, under the conditions used, there was, on average, slightly stronger binding with ITE than with FICZ.

Canonical HIF binding sites were identified in a similar way, using peaks that had either HIF-1α or HIF-2α signal together with HIF-1β/ARNT signal, in both replicates of a condition. In total, 983 canonical HIF binding sites were identified in hypoxic cells treated with DMSO (828 HIF-1 sites and 401 HIF-2 sites, of which 246 were present in both datasets) (Fig 3B). 101 of these sites were also detected in

normoxia, suggesting low-level HIF activation under this condition. Consistent with this, no sites were detected in normoxia alone. As expected, HIF-1α and HIF-2α binding was strongly induced by hypoxia at canonical HIF-1 and HIF-2 sites, respectively, with concomitant increases also observed in HIF-1β/ARNT binding (Fig S5N–X). Structural analysis has shown that the HIF-1β/ARNT subunit of HIF and AHR binds a GTG DNA sequence (33). Consistent with this, both AHR and HIF binding sites were strongly enriched for a consensus NCGTG motif (Fig 3C–E). However, HIF-1 and HIF-2 strongly favoured the ACGTG variant (the hypoxia response element), whereas AHR preferentially recognised the GCGTG motif (the xenobiotic/dioxin response element).

We next investigated the interaction between HIF and AHR activation on the binding of the opposing factor to chromatin. Hypoxia significantly reduced binding of both AHR and HIF-1β/ARNT at canonical AHR sites in ITE-treated cells (Figs 3F–I and S6A–C). This was independent of signal intensity at the site. Hypoxia also had a weaker effect on AHR and HIF-1β/ARNT binding in FICZ-treated cells (Fig S6D–G). Taken together, these results indicate that hypoxia reduces AHR DNA binding in RCC4+VHL cells. Conversely, ITE treatment significantly reduced HIF-1α binding at canonical HIF-1 sites and HIF-2α binding at canonical HIF-2 sites with concomitant reductions in HIF-1β/ARNT binding (Figs 3J–Q and S6H–J). Again, weaker effects were seen when the cells were treated with FICZ (Fig S6K–R). Overall, AHR activation reduces both HIF-1 and HIF-2 binding in RCC4+VHL cells. Thus, we observe bi-directional competition between HIF-α and AHR at the level of chromatin binding in RCC4+VHL cells that reflects their bi-directional competition for HIF-1β/ARNT.

In clear cell renal cancer, HIF-α is constitutively stabilised irrespective of oxygen levels because of VHL inactivation. Because HIF activated in this way may differ from HIF stabilised by hypoxia, we examined the effect of this normoxic HIF activation on binding of AHR and HIF-1β/ARNT to canonical AHR sites by comparing signals in VHL WT RCC4+VHL cells and VHL-deficient RCC4+VA cells in normoxia (Fig S7). VHL loss led to a significant reduction in both AHR and HIF-1β/ARNT binding at canonical AHR sites in cells treated with ITE, comparable to that observed under hypoxia (Fig S7A–D). Conversely, in VHL-defective RCC4+VA cells, ITE significantly reduced binding of HIF-1α, HIF-2α, and HIF-1β/ARNT at canonical HIF binding sites (Fig S7E–L). Taken together, this indicates that the crosstalk between AHR and HIF on chromatin binding is a direct consequence of HIF activation rather than an off-target effect of hypoxia. It also confirms the relevance of our findings to kidney cancer, indicating that HIF activation by VHL loss can alter AHR binding and that AHR stimulation can reciprocally modulate the HIF response.

### Reduced competition between HIF and AHR at shared chromatin binding sites

Of the 1,933 canonical HIF and AHR binding sites identified, 221 (11%) were overlapping HIF and AHR binding sites. This overlap occurred at 19% of all AHR sites and was higher at HIF-2 sites (42% of sites) than at HIF-1 sites (18% of sites) (Fig 4A). Of note, canonical AHR binding sites also showed a similar promoter-distal distribution to canonical HIF-2 binding sites in contrast to the more promoter-

proximal distribution observed for canonical HIF-1 binding sites (Fig 4B–D). This similarity in distribution may in part explain the greater overlap between AHR and HIF-2 binding sites.

The hypoxia response element HIF-recognition motif (RCGTG) and the xenobiotic/dioxin response element AHR-recognition motif (GCGTG) are very similar. This raises the possibility that in addition to a global competition for HIF-1β/ARNT, HIF and AHR may compete for access to the same binding motif on chromatin at these overlapping sites. To dissect this further, we examined whether shared AHR and HIF sites were subjected to a different level of competition than sites that bound either transcription factor alone. Contrary to our hypothesis, hypoxia suppressed AHR binding at shared sites less than it did at sites that bound AHR alone. Specifically, the hypoxic repression of AHR binding at sites shared with HIF-1 (median $\log_2$FC: −1.199) or HIF-2 (median $\log_2$FC: −1.264) was significantly weaker than at individual sites (median $\log_2$FC: −1.458) (Fig 4E and F). Conversely, the ITE-dependent reduction in HIF-1α binding was also weaker at sites shared with AHR (median $\log_2$FC: −0.490) than at individual sites (median $\log_2$FC: −0.591) (Fig 4G). The difference in repression of HIF-2α binding at shared sites (median $\log_2$FC: −0.697) versus individual sites (median $\log_2$FC: −0.703) did not reach statistical significance (Fig 4H). Taken together, this suggests that rather than competing for the same binding motif, HIF and AHR positively reinforce chromatin binding of each other in cis when bound at overlapping sites to an extent that partially mitigates the competition for HIF-1β/ARNT.

To gain better insight into the potential mechanisms underlying this, we attempted to resolve the precise binding overlap between these factors at shared sites by comparing the peak summits. As a control, we established that the median distance between summits for AHR ChIP-seq signal when stimulated with either ITE or FICZ was just 19 bp. However, the median distance between AHR and HIF-1α summits was significantly larger at 39 and 38 bp (for ITE and FICZ, respectively), whereas the distance between AHR and HIF-2α summits was 62 and 59 bp (Fig 4I and J). Taken together, this suggests that even where AHR and HIF-α ChIP-seq peaks overlap, the two transcription factors generally bind to closely associated but distinct sites (Fig S8, e.g., loci), consistent with HIF and AHR binding to the two closely related but largely distinct recognition motifs ACGTG and GCGTG, respectively.

### Contrasting antagonistic and cooperative activities of HIF and AHR in gene regulation

The observation of both general and locus-specific effects in the interaction between HIF and AHR transcription pathways suggested that there might be heterogenous consequences on target gene expression that go beyond general antagonism. Therefore, to relate changes in the DNA binding of HIF and AHR across the genome to downstream changes in gene expression, we performed RNA-sequencing on RCC4+VHL cells incubated in 0.5% hypoxia and/or treated with ITE for 6 h.

Gene set enrichment analysis (GSEA) revealed that AHR-bound genes—those closest to canonical AHR-only binding sites—were enriched among ITE-up-regulated genes (enrichment score [ES]: 0.98; P-value = 0.03) and also enriched among hypoxia-down-regulated genes in cells treated with ITE (ES: −0.41; P-value = 1 ×

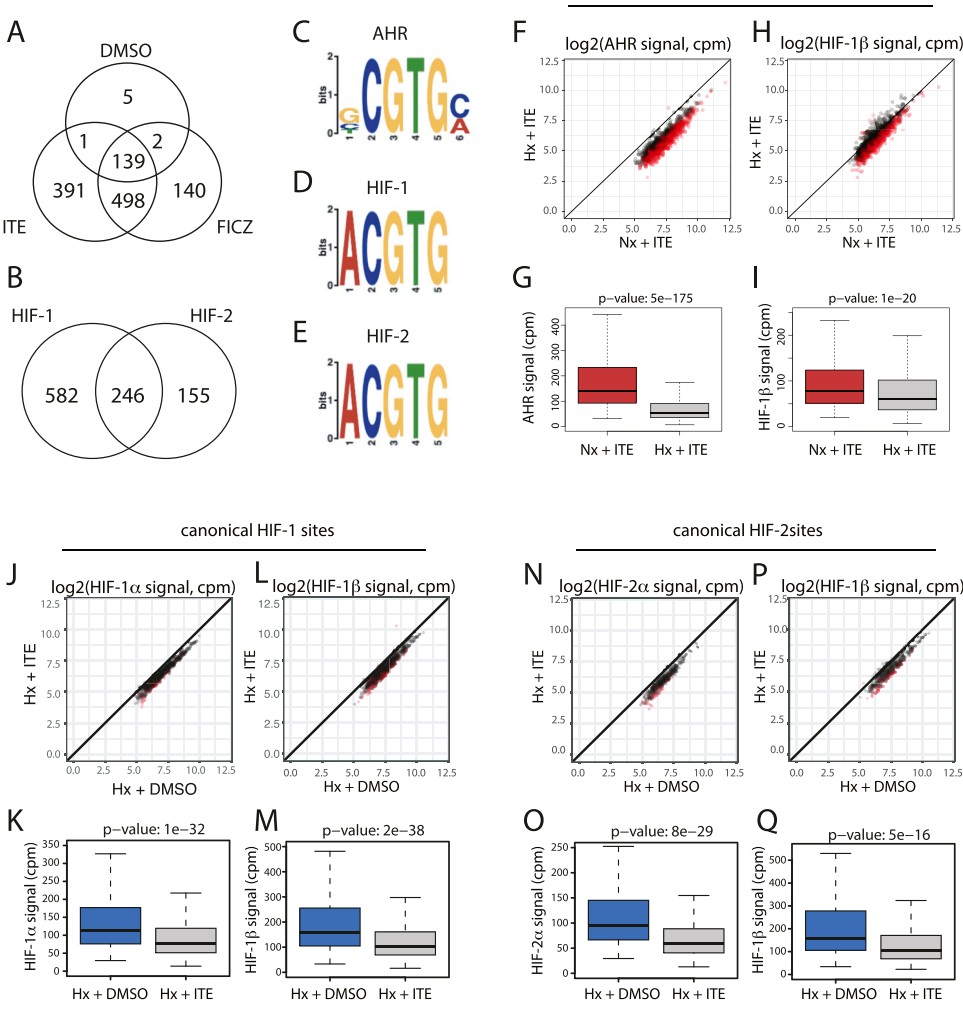

**Figure 3. HIF and AHR inhibit binding of each other to chromatin in trans.**
**(A)** Canonical AHR binding sites (those binding both AHR and HIF-1β/ARNT in both ChIP-seq replicates) were identified in RCC4+VHL cells treated with DMSO (control), ITE, or FICZ. The Venn diagram shows sites detected under the different stimuli. **(B)** Canonical HIF-1 binding sites in hypoxic RCC4+VHL cells (binding both HIF-1α and HIF-1β/ARNT in both biological replicates) and canonical HIF-2 binding sites (binding both HIF-2α and HIF-1β/ARNT in both biological replicates) were identified. The Venn diagram shows the number of sites bound by one or both factors. **(C, D, E)** Most significantly enriched sequence logos in DREME analysis (32) of (C) canonical AHR unique sites, (D) canonical HIF-1 unique sites, and (E) canonical HIF-2 unique sites. **(F, G)** Scatter plot and (G) box-and-whisker plot showing normalised AHR signal intensity at canonical AHR binding sites (averaged across the two replicates) in RCC4+VHL cells treated with hypoxia + ITE versus normoxia + ITE. **(H, I)** The same analysis for HIF-1β/ARNT signal at canonical AHR sites. **(J, K)** Scatter plot and (K) box-and-whisker plot showing normalised HIF-1α signal intensity at canonical HIF-1 binding sites (averaged across the two replicates) in RCC4+VHL cells treated with hypoxia + DMSO versus hypoxia + ITE. **(L, M)** The same analysis for HIF-1β/ARNT signal at canonical HIF-1 sites. **(N, O)** Scatter plot and (O) box-and-whisker plot showing normalised HIF-2α signal intensity at canonical HIF-2 binding sites (averaged across the two replicates) in RCC4+VHL cells treated with hypoxia + DMSO versus hypoxia + ITE. **(P, Q)** The same analysis for HIF-1β/ARNT signal at canonical HIF-2 sites. Data information: P-values in box-and-whisker plots (G, I, K, M, O, Q) by Wald test with correction using Benjamini–Hochberg procedure.

$10^{-04}$) (Fig 5A and B). This is consistent with transcriptional activation by AHR that is inhibited in hypoxia and consistent with the overall changes in AHR chromatin binding seen by ChIP-seq. Conversely, HIF-bound genes—those closest to canonical HIF-only binding sites—were enriched among hypoxia-up-regulated genes (ES: 0.97; P-value = 1 × $10^{-04}$), although they were not significantly enriched among ITE-down-regulated genes under hypoxia (ES: −0.21; P-value = 0.25) (Fig 5C and D). Furthermore, AHR-specific target genes that were up-regulated by ITE and bound by AHR, but not by HIF, were down-regulated in hypoxia (median $log_2FC$: −0.235; P-value = 3 × $10^{-06}$, compared with normoxia), whereas HIF-specific target genes that were up-regulated by hypoxia and bound by HIF, but not by AHR, were down-regulated by ITE (median $log_2FC$: −0.076; P-value = 0.009, compared with DMSO) (Fig 5E). This bi-directional nature of the crosstalk between HIF and AHR in gene regulation is consistent with that observed for chromatin binding.

In contrast, co-bound genes—those closest to both an AHR and HIF binding site—were significantly enriched among both hypoxia-up-regulated (ES: 0.97; P-value = 0.03) and ITE-up-regulated genes (ES: 0.96; P-value = 0.07), consistent with these genes being direct

transcriptional targets of both transcription factors (Fig 5F and G). This analysis defined a small set of 14 genes (*PAX2*, *MOAP1*, *WSB1*, *FAM65C*, *MFNG*, *CARD10*, *LIMCH1*, *NR3C1*, *PADI1*, *SLC2A1*, *ARNTL*, *INSIG1*, *PAG1*, *VLDLR*) that are bound by both HIF and AHR and up-regulated by both hypoxia and ITE. For these genes, there was an additive effect on gene regulation when the two stimuli were combined (Fig 5H). This characterises a third point of interaction between the two transcriptional pathways, in which a small number of genes are directly regulated by both transcription factors.

## Evidence of HIF and AHR antagonism in ccRCC tumours

To further evaluate the relevance of our findings to kidney cancer, we next examined the crosstalk between HIF and AHR in RNA-sequencing analyses of ccRCC tumours from The Cancer Genome Atlas (TCGA-KIRC) cohort. Because HIF-α and AHR mRNA levels do not represent activation of these pathways, we first used our ChIP-seq and RNA-seq analyses from RCC4+VHL cells to define an AHR gene signature comprising 67 AHR-bound and ITE-up-regulated genes and an HIF gene signature comprising 236 HIF-bound and hypoxia-induced genes (genes bound and up-regulated by both

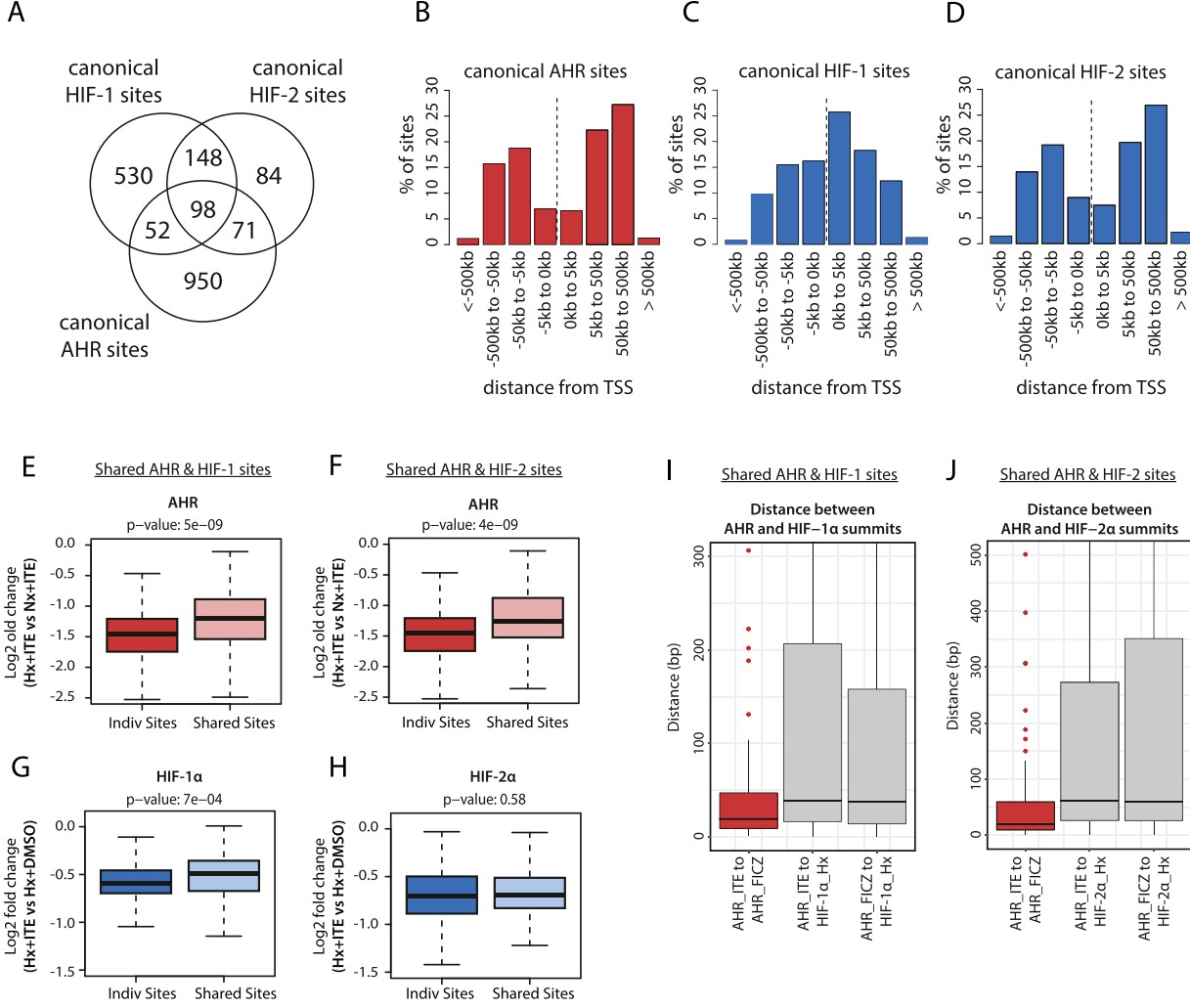

**Figure 4. HIF and AHR promote binding of each other to chromatin in cis.**
**(A)** Venn diagram showing the number of individual and overlapping canonical HIF-1, HIF-2, and AHR binding sites. **(B, C, D)** Bar charts showing the frequency distribution for (B) canonical AHR sites, (C) canonical HIF-1 sites, and (D) canonical HIF-2 sites with respect to the distance upstream or downstream from the closest transcriptional start site. **(E)** Box-and-whisker plot showing the $\log_2$ fold change in AHR signal intensity between hypoxia + ITE and normoxia + ITE for canonical AHR binding sites without HIF binding (Indiv Sites) and with HIF-1 binding (Shared Sites). **(F)** The same analysis for canonical AHR sites without HIF binding (Indiv Sites) and with HIF-2 binding (Shared Sites). **(G)** Box-and-whisker plot showing the $\log_2$ fold change in HIF-$\alpha$ signal intensity between hypoxia + ITE and hypoxia + DMSO for canonical HIF-1 binding sites without AHR binding (Indiv Sites) and with AHR binding (Shared Sites). **(H)** The same analysis for canonical HIF-2 sites without AHR binding (Indiv Sites) and with AHR binding (Shared Sites). **(I, J)** Box-and-whisker plots showing the distance between (I) AHR and HIF-1$\alpha$ and (J) AHR and HIF-2$\alpha$ peak summits (grey boxes). For comparison, the distance between AHR peak summits when stimulated with either ITE or FICZ is also shown (red boxes). Data information: $P$-values in box-and-whisker plots (E, F, G, H) by Wald test with correction using Benjamini–Hochberg procedure.

factors were excluded). After normalisation, individual gene expression levels were combined to give composite scores reflecting overall AHR or HIF activation. We then compared these composite scores for clear cell kidney tumours and paired surrounding kidney tissue. As expected, HIF composite scores were significantly elevated in ccRCC tumours compared with patient-matched normal kidney ($P = 6 \times 10^{-13}$), confirming constitutive HIF activation in these tumours as a result of VHL loss (Fig 6A). In contrast, AHR composite scores were significantly lower in tumours compared with patient-matched normal kidney ($P = 1 \times 10^{-5}$) (Fig 6B). Importantly, when we repeated the analysis using a composite score based on common HIF- and AHR-target genes, no significant difference was observed, indicating that the increase in HIF activity is balanced by the

reduction in AHR activity at these common loci (Fig 6C). In ccRCC, VHL inactivation frequently results (at least in part) from copy number loss. In the TCGA-KIRC tumour samples, increasing VHL copy number loss was associated with a significant increase in the HIF composite score ($P = 7 \times 10^{-6}$) (Fig 6D), whereas an inverse association was seen for the AHR composite score ($P = 5 \times 10^{-11}$) (Fig 6E). Conversely, no significant association was observed for a composite score based on the common HIF- and AHR-target genes (Fig 6F). Finally, we compared the HIF and AHR composite scores for each individual tumour and found a significant negative correlation between the two composite scores, further supporting an antagonism between the pathways (Fig 6G). Taken together, we observe a negative correlation between overall HIF and AHR transcriptional

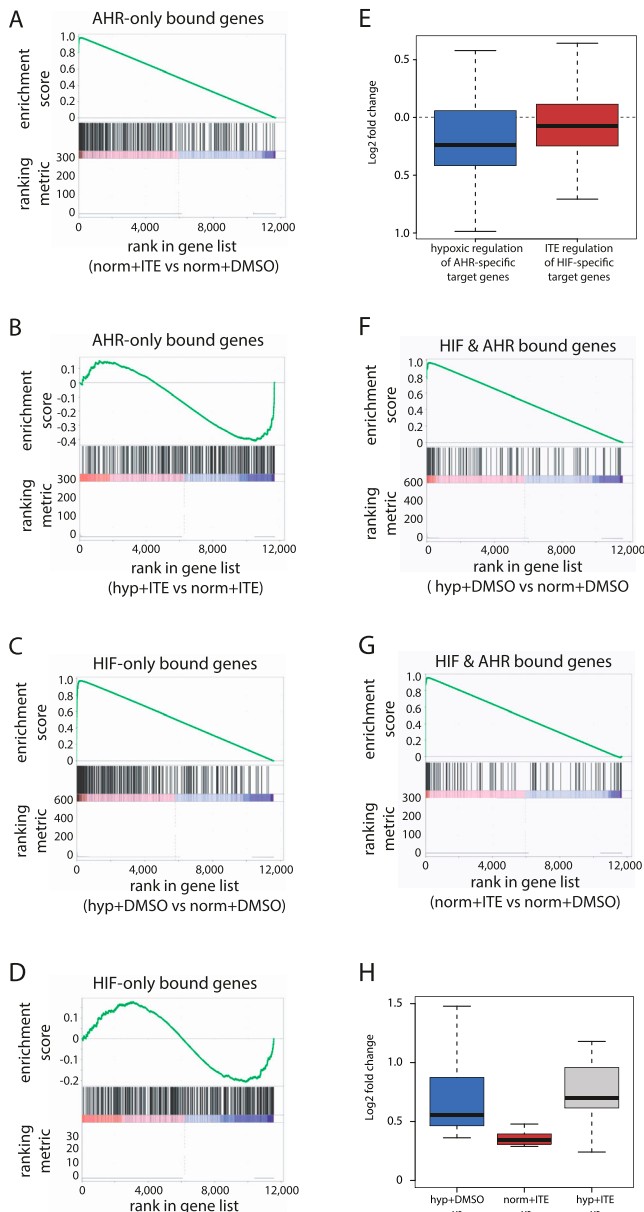

**Figure 5. Heterogenous effects of crosstalk between HIF and AHR on gene regulation.**
**(A)** Gene set enrichment analysis (GSEA) showing enrichment of genes adjacent to canonical AHR binding sites among genes induced in normoxia + ITE compared with normoxia + DMSO (ranked on the x-axis). **(B)** GSEA showing enrichment of the same genes among genes down-regulated in hypoxia + ITE compared with normoxia + ITE. **(C)** GSEA showing enrichment of genes adjacent to canonical HIF binding sites among genes up-regulated in hypoxia + DMSO compared with normoxia + DMSO. **(D)** GSEA showing enrichment of the same genes among genes down-regulated in hypoxia + ITE compared with hypoxia + DMSO. **(E)** Box-and-whisker plot showing down-regulation of AHR-induced genes in hypoxia + ITE compared with normoxia + ITE (blue; P-value = 3 × 10$^{-06}$; paired t-test) and of HIF-bound only and hypoxia-induced genes in hypoxia + ITE compared with hypoxia + DMSO (red; P-value = 0.009; paired t-test). **(F)** GSEA showing enrichment of HIF and AHR co-bound genes among genes up-regulated in hypoxia + DMSO compared with normoxia + DMSO. **(G)** GSEA showing enrichment of HIF and AHR co-bound genes among genes up-regulated in normoxia + ITE compared with normoxia + DMSO. **(H)** Box-and-whisker plot showing the regulation by hypoxia (blue), ITE (red), or the combination of both stimuli (grey) of common HIF- and AHR-target genes.

activation in ccRCC tumours from the TCGA-KIRC cohort. This is consistent with the antagonism seen between the two transcription factors in ccRCC cell lines and indicates that endogenous AHR activity in ccRCC interacts with HIF activation.

Finally, the HIF pathway has long been considered a target for the treatment of ccRCC tumours. In this tumour type, HIF-2 has a role as a key ccRCC oncogene, whereas HIF-1 appears to act more as a tumour suppressor and is often repressed by the tumour. Thus, effort has focussed on the development of small molecule HIF-2 dimerization inhibitors and their use as therapeutic agents (34). The first HIF-2 dimerization inhibitor (belzutifan/PT2977) has recently been approved by the US Food and Drug Administration for the treatment of ccRCC in patients with the familial VHL syndrome (14, 35). Because blocking HIF-2 dimerization will liberate HIF-1β/ARNT, we next evaluated the effect of the first-generation HIF-2 inhibitor PT2385 on AHR activity. 786O parental cells (defective for both WT VHL and HIF-1α) were pre-treated with PT2385 for 18 h (to ensure complete inhibition of constitutive HIF-2 activity and downstream gene expression) followed by ITE for 18 h. PT2385 treatment completely abolished expression of the HIF-specific gene *NDRG1*, which was also significantly repressed by ITE treatments (Fig 6H). Conversely, PT2385 treatment significantly increased the ITE-dependent induction of the AHR-specific genes *CYP1B1* and *ALDH1A3*. Taken together, this indicates that in addition to direct effects on HIF-2 signaling in ccRCC tumours, belzutifan may have indirect effects on AHR activation. Because AHR activation may promote tumourigenesis, this raises the possibility that there may be therapeutic synergy in targeting both pathways simultaneously.

# Discussion

In this work, we demonstrate diverse and unexpected interactions between the HIF and AHR pathways that operate both generally and in a locus-specific manner to influence chromatin binding and gene expression. These interactions are both antagonistic and cooperative and include (1) competition between HIF-α and AHR subunits for binding to HIF-1β/ARNT, (2) synergistic binding, in cis, of the two transcription factors when bound close to each other on chromatin, and (3) partial overlap in the transcriptional targets of the two transcription factors. Our findings provide new insights into both the nature of the interplay between the HIF and AHR pathways and its implications for their therapeutic manipulation in cancer.

## Mechanistic implications

Firstly, we have shown that AHR and HIF-α compete for binding to HIF-1β/ARNT, leading to global antagonism between the two pathways. Previous studies have shown that this can be reversed by overexpressing HIF-1β/ARNT, confirming that it is a limiting subunit (36). Also, consistent with previous reports (28), activation of HIF inhibited AHR-target gene expression and AHR-HIF-1β/ARNT dimerization in all settings, but there was no significant effect of AHR activation on HIF-target genes in three out of five cell lines studied. Only in two ccRCC cell lines, 786O + VHL and RCC4+VHL, did activation of AHR inhibit HIF-target gene expression and only in

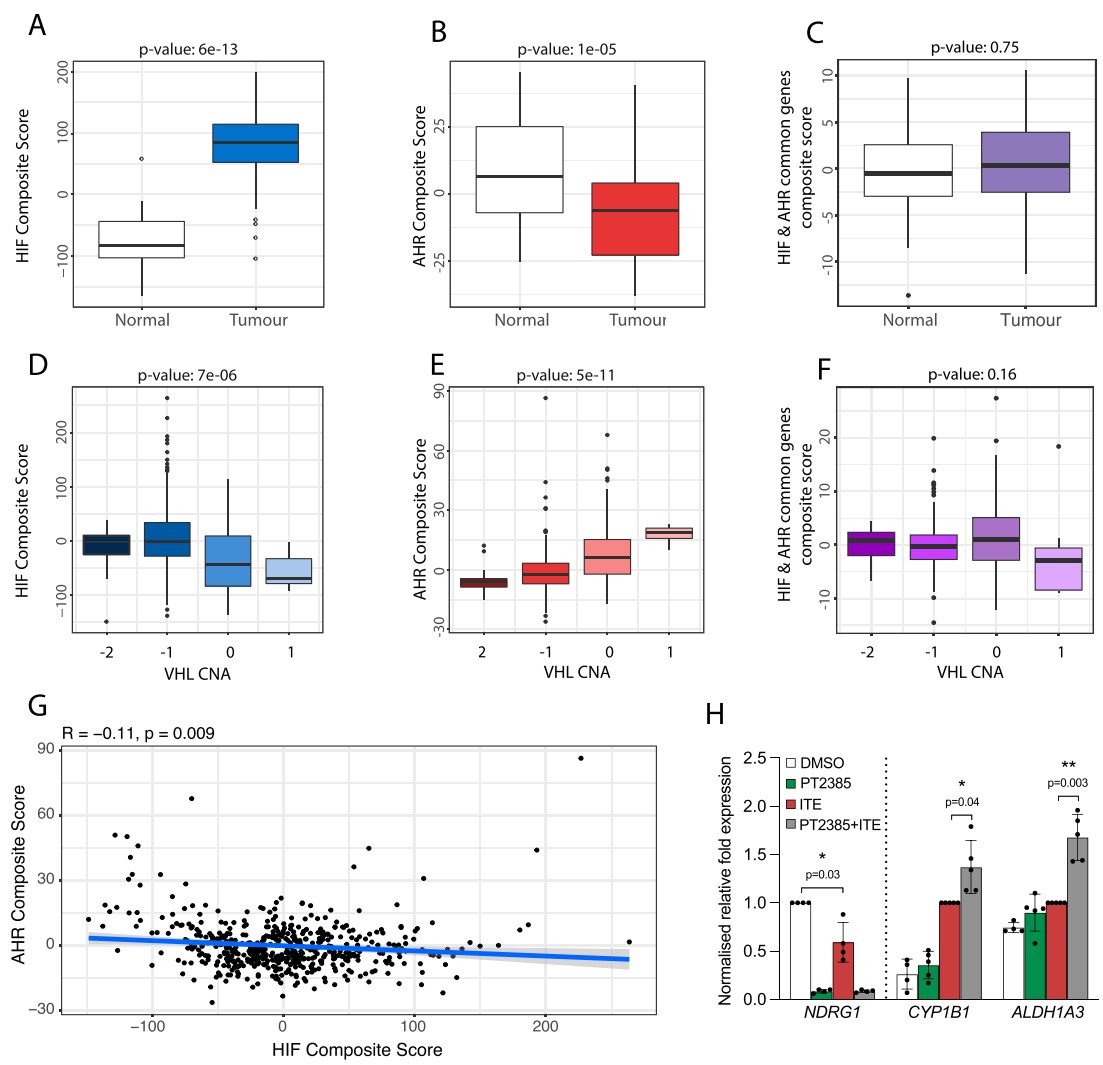

**Figure 6. Crosstalk between HIF and AHR in ccRCC tumours and in response to chemotherapeutic agents.**
**(A)** Box-and-whisker plot showing the composite score for an HIF metagene (based on 236 genes that were bound only by HIF and up-regulated by hypoxia in DMSO-treated RCC4+VHL cells) in RNA-seq analysis of normal and tumour samples from the TCGA-KIRC cohort of clear cell renal cancer. **(B, C)** The same analysis for (B) an AHR composite score based on 67 genes that were bound only by AHR and up-regulated by ITE in normoxic RCC4+VHL cells and (C) a composite score based on common HIF- and AHR-bound and regulated genes. **(D, E, F)** The same analyses performed on tumour samples stratified according to copy number alteration at the *VHL* gene locus. **(G)** Scatter plot showing the correlation between AHR and HIF composite scores for the TCGA-KIRC tumour samples. *P*-values (A, B, C, D, E, F, G) by paired Wilcoxon rank sum test. **(H)** RT-qPCR analysis of *NDRG1*, *CYP1B1*, and *ALDH1A3* mRNA levels in 786O cells exposed to DMSO (control), PT2385, ITE, or both, presented as mean ± SD of five biological replicates. *P ≤ 0.05, **P ≤ 0.01 (paired *t*-test).

RCC4+VHL cells did the crosstalk between the two factors operate in both directions. Notably, these two cell lines had relatively high ratios of AHR to HIF-α subunits. Therefore, although the relative affinities of HIF-1α, HIF-2α, and AHR for HIF-1β/ARNT may influence the symmetry and strength of their competition (37), the variation in behaviour between cell lines and its broad correlation with the relative stoichiometry of the different subunits suggests that the latter also has a role in defining the directionality of this competition for HIF-1β/ARNT. In this respect, it is also important to consider other β-class bHLH-PAS paralogues of HIF-1β/ARNT: ARNT2, ARNTL, and ARNTL2 (1, 2, 3). Interaction of AHR and HIF-α with ARNTL and ARNTL2 is relatively weak and therefore unlikely to be of significance in vivo (3, 38, 39, 40). However, studies suggest that ARNT2 can substitute for HIF-1β/ARNT to dimerize with both AHR

and HIF-α subunits and direct transcription (3, 38, 39, 40). Nevertheless, *ARNT2* expression is highly tissue specific, being predominantly expressed in the brain and so is unlikely to contribute to the crosstalk between AHR and HIF-α outside of restricted cell types (40). Indeed, analysis of publicly available RNA-seq analyses indicates that *ARNT* mRNA is 7–35 times more abundant than *ARNT2* mRNA in the cell lines studied.

The limited availability of HIF-1β/ARNT raises the possibility of even more complex interplay between HIF, AHR, and other bHLH-PAS family members that are also known to dimerize with HIF-1β/ARNT. Specifically, AHRR (aryl hydrocarbon receptor repressor), the SIM proteins (single-minded 1 and 2), and NPAS1 and NPAS3 (neuronal PAS domain protein 1 and 3) have all been reported to dimerize with HIF-1β/ARNT (41, 42, 43). Notably, a competitive

relationship between AHR and AHRR has previous been described, whereby AHRR was found to sequester HIF-1β/ARNT and reduce AHR transcriptional activity (43, 44). As a consequence, AHR was redirected towards a non-canonical cytoplasmic E3 ligase function, targeting the estrogen receptor α for degradation (44). The availability of HIF-1β/ARNT was therefore suggested as the basis for a switch between AHR canonical and non-canonical functions, and notably, we have observed relocation of AHR from nuclear to cytoplasmic cell fractions in response to HIF-α stabilisation (Figs 2A and B and S3A and B). Whether reduced HIF-1β/ARNT availability under HIF-stabilising conditions is able to redirect AHR to other non-canonical cytoplasmic functions in RCC cells requires further investigation but suggests an additional level of interplay between HIF and the AHR pathway.

Secondly, analysis of AHR and HIF DNA-binding across the genome revealed several further levels of interaction. In RCC4+VHL cells, the competition between AHR and HIF-α subunits for HIF-1β/ARNT was reflected in binding to chromatin, with activation of either HIF or AHR leading to a reciprocal global reduction in binding of the other factor. This is likely a direct consequence of competition for HIF-1β/ARNT. However, where HIF and AHR bound close to each other, this mutual inhibition was reduced. This indicates that rather than competing for binding to shared chromatin motifs, HIF and AHR positively reinforce the binding of each other in cis. Indeed, when bound at overlapping sites, AHR and HIF binding do not coincide precisely, indicating that they bind to separate response elements. Furthermore, cooperation between transcription factors is commonly observed at enhancers and can result through a direct interaction between simultaneously co-bound factors or indirectly through effects on chromatin remodelling or nucleosome positioning (45, 46).

Thirdly, a small set of genes that are bound by both transcription factors are also induced by each factor independently and in combination, indicating that they are direct transcriptional targets of both AHR and HIF, which can act synergistically. Notably, this list of genes includes *ARNTL*, which is the dimerization partner for the regulator of circadian rhythms, CLOCK (38). This raises the possibility of still further crosstalk between AHR, HIF, and other bHLH-PAS transcription factors.

**Relevance to kidney cancer**

AHR mRNA levels are elevated in many types of cancer when compared with normal tissues (including ccRCC), and AHR has important roles in regulating cellular proliferation, invasion, migration, and immunity that help drive tumour initiation, promotion, progression, and metastasis (26, 47, 48, 49). However, AHR activation can have both pro- and anti-tumourigenic actions depending upon the cell line or tumour model used. In ccRCC, activation of AHR increases cancer cell invasion and is associated with poor patient survival (30). Conversely, pharmacological stimulation of AHR may also slow proliferation and migration of ccRCC cells in vitro and in xenograft models (50).

Similarly, HIF is activated in many cancer types, in which it is associated with intra-tumour hypoxia, poor patient prognosis, and resistance to therapy. However, like AHR, HIF can be both pro- and anti-tumourigenic depending upon the setting, and in ccRCC, HIF-2

promotes tumour growth, whereas HIF-1 is tumour suppressive. This has led to the development of selective HIF-2 inhibitors, which specifically block dimerization of HIF-2α with HIF-1β/ARNT to slow renal cancer progression (34). However, the new data reveal that in ccRCC tumours, competition for HIF-1β/ARNT results in a reciprocal relationship between HIF and AHR and constitutive activation of HIF is associated with suppression of the AHR pathway. Therefore, therapies that block HIF-2 dimerization can release more HIF-1β/ARNT to bind with AHR, thereby potentially promoting the oncogenic effects of the AHR pathway in kidney cancer. In summary, our data predict both that therapeutic inhibition of AHR in RCC has activating effects on components of the HIF pathway and that inhibition of HIF has activating effects on the AHR pathway. This raises important questions as to whether combination therapy targeting both pathways might show synergistic efficacy over and above targeting each system separately.

# Materials and Methods

### Cell culture

RCC4 renal cancer cells were a gift from CH Buys. RCC4+VHL and RCC4+VA ("vector alone") clones were generated by stable transfection with pcDNA3-VHL or pcDNA3, respectively (9). 786O renal cancer cells, HepG2 liver cancer cells, and PC3 prostate cancer cells were purchased directly from ATCC. The 786O+VHL clone was a gift from WG Kaelin. HKC8 renal tubular cells were a gift from L.C. Racusen (51). HKC8 HIF-1α/KO, HIF-2α/KO, and 1α/2α dKO clones were generated through CRISPR-Cas9 disruption, as previously described (13). Unless used directly from a certified source, the identity of all cell lines was confirmed by STR genotyping. RNA-seq analysis of RCC4 and 786O cells also confirmed the presence of unique VHL gene coding mutations (chr3:10,183,725C > G and chr3:10,183,841 G > del, respectively) that are as previously described. All cell lines were grown in DMEM supplemented with 100 U/ml penicillin, 100 μg/ml streptomycin, 2 mM L-glutamine, and 10% foetal bovine serum (Sigma-Aldrich) and regularly tested for mycoplasma infection. RCC4+VHL, RCC4+VA and 786O+VHL cells were maintained in 0.5 mg/ml G418. Cells were maintained at 37°C under an atmosphere of 5% $CO_2$ in air. Cells were allowed to grow in phenol red–free DMEM supplemented with 100 U/ml penicillin, 100 μg/ml streptomycin, 2 mM L-glutamine, and 10% dextran-coated charcoal-treated foetal calf serum for ~48 h before treatments to minimise basal AHR activity. Treatments of sub-confluent cells were performed as indicated with 100 nM ITE, 10 nM FICZ (both from Tocris Bioscience), 50 μM FG4592 (Selleckchem), 1 μM PT2385 (MedChemExpress), or 0.1% DMSO for specified duration. Hypoxic incubations were conducted for specified duration at 0.5% ambient oxygen concentration (5% $CO_2$; balance $N_2$) within an InvivO2 400 workstation (Baker Ruskinn).

### Whole cell lysate preparation

Cells were washed with PBS, and lysates were prepared in urea-SDS lysis buffer (6.7 M urea, 10% glycerol, 1% SDS, 10 mM Tris–HCl, pH 8.0)

supplemented with freshly added cOmplete Protease Inhibitor Cocktail (Roche) and 1 mM DTT. Lysates were incubated 15 min at 4°C, then sonicated using a Bioruptor (Diagenode) at medium intensity for four cycles of 15 s on/30 s off, before being centrifuged for 5 min at 16,000$g$ at 4°C. Protein concentrations were determined by BCA assay (Pierce).

## Cytoplasmic and nuclear lysate preparation

Cytoplasmic and nuclear protein extraction was performed based on the published Method 2 from Klenova et al ([52]). Briefly, cells were washed with PBS and resuspended in Tween-20 lysis buffer (0.5% Tween-20; 25 mM Tris–HCl, pH 8.0; 2 mM EDTA, pH 8.0) containing 50 mM NaCl and supplemented with freshly added cOmplete Protease Inhibitor Cocktail (Roche). Lysates were incubated for 15 min at 4°C, then centrifuged for 1 min at 6,000$g$ at 4°C. Supernatants were recovered as cytoplasmic fractions. Nuclear pellets were washed and resuspended in Tween-20 lysis buffer containing 500 mM NaCl. Lysates were incubated 15 min at 4°C before, then supplemented with Tween-20 lysis buffer without NaCl (for a final concentration of 250 mM) and centrifuged for 15 min at 10,000$g$ at 4°C. Supernatants were kept as nuclear fractions. Protein concentrations were determined by BCA assay (Pierce).

## Immunoprecipitation

Cytoplasmic or nuclear cell lysates (between 150–300 $\mu$g per experiment) were diluted to 1 ml in Tween-20 buffer (0.5% Tween-20; 25 mM Tris–HCl, pH 8.0; 2 mM EDTA, pH 8.0) containing 150 mM NaCl and supplemented with freshly added cOmplete Protease Inhibitor Cocktail (Roche). Samples were pre-cleared by incubating with protein A-agarose beads (Millipore) for 1 h at 4°C on a rotating mixer. Samples were then centrifuged for 3 min at 400$g$. Supernatants were incubated with non-immunized rabbit serum (X0902; Dako) or HIF-1$\beta$/ARNT antibody (rabbit polyclonal; cat no. NB100-110; Novus Biologicals) (between 2–4 $\mu$l) overnight at 4°C on a rotating mixer, before incubation with protein A-agarose beads (between 25–50 $\mu$l) for 1 h at 4°C. Beads were washed three times in Tween-20 buffer for 5 min at 4°C. Protein complexes were eluted from beads by adding Laemmli sample buffer with 50 mM DTT and incubating at 95°C for 5 min. Proteins of interest were detected by immunoblotting IP and input samples (HIF-1$\beta$/ARNT was detected using mouse monoclonal antibody NB100-124; see below).

## Immunoblotting

Samples were prepared for SDS–PAGE by adding Laemmli sample buffer with 50 mM DTT and incubating at 95°C for 5 min. Proteins were then resolved on acrylamide gels using Mini-PROTEAN Tetra cells (Bio-Rad) and transferred to polyvinylidene difluoride membrane (Immobilon-P; Millipore) using mini-trans-blot cells (Bio-Rad). Membranes were blocked for 1 h in 5% milk solution prepared in PBS-T (phosphate-buffered saline containing 0.1% Tween-20), before being incubated in primary antibodies overnight at 4°C (in 5% milk/PBS-T). Primary antibodies used were anti-AHR (A-3, cat no. sc-133088; Santa Cruz Biotechnology), anti-HIF-1$\alpha$ (cat no. 610959; BD Biosciences), anti-HIF-2$\alpha$ (mouse monoclonal, 190 b),

anti-HIF-1$\beta$/ARNT (D28F3, cat no. 5537; Cell Signaling), anti-HIF-1$\beta$/ARNT (H1$\beta$234, cat no. NB100-124; Novus Biologicals), anti-CYP1B1 (EPR14972, cat no. ab185954; Abcam), anti-NDRG1 (cat no. 5196; Cell Signaling), anti-HDAC2 (cat no. PA1-861; Invitrogen), anti-$\alpha$-tubulin (cat no. ab4074; Abcam), anti-PolII (CTD4H8, cat no. 05-623; Millipore), and anti-$\beta$-actin-HRP (cat no. ab49900; Abcam). HRP-conjugated secondary antibodies (rabbit, cat no. P0048; mouse, cat no. P0047; Dako) were incubated for 1 h at room temperature. Membranes were developed using either SuperSignal West Dura Extended Duration Substrate or SuperSignal West Femto Maximum Sensitivity Substrate (Thermo Fisher Scientific) and imaged using a ChemiDoc XRS+ imaging system (Bio-Rad). After immunoblot analysis, membranes were stained with Coomassie brilliant blue to visualize separated proteins. Densitometric analysis was performed using Image Lab Software (Bio-Rad) and expressed as the mean ± SD of at least three independent experiments. Statistical significance was tested by paired $t$-test using GraphPad Prism software (v9).

## Chromatin immunoprecipitation (ChIP) coupled to qPCR

ChIP experiments were performed as previously described ([53]) using antibodies directed against AHR (rabbit monoclonal; D5S6H, cat no. 83200; Cell Signaling), HIF-1$\alpha$ (rabbit polyclonal, PM14), HIF-2$\alpha$ (rabbit polyclonal, PM9), or HIF-1$\beta$/ARNT (rabbit polyclonal; cat no. NB100-110; Novus Biologicals) ([54], [55]). Non-immunized rabbit serum (X0902; Dako) was used as a negative control. One 15-cm plate of cells was used for each ChIP. qPCR analysis was performed using Fast SYBR Green Master Mix on a StepOne thermocycler (both from Applied Biosystems) with the following oligonucleotides: HIF binding site at the *NDRG1* locus (forward: TCCCTCCCAATCTCTCTCTTCTT; reverse: CACCATCAGCACAGCAAACTAC) and AHR binding site at *CYP1B1* locus (forward: CGCTCCGGCGAACTTTAT; reverse: GATATGACTGGAGCCGACTTTC).

## RNA extraction and RT–qPCR

Cells were lysed in TRIzol and total RNA was prepared using Direct-zol RNA MiniPrep Kit (Zymo Research), according to manufacturer's instructions. Equal amounts of RNA were used for cDNA synthesis using the High-Capacity cDNA Reverse Transcription Kit (Applied Biosystems). Expression analyses were performed using Fast SYBR Green Master Mix on a StepOne thermocycler (both from Applied Biosystems, now Thermo Fisher Scientific) and calculated using the Pfaffl method. The list of primers used for analysis is found in Table S1, with *HPRT1* used as reference gene for normalisation. Data are expressed as the mean ± SD of at least three independent experiments. Statistical significance was tested by paired $t$-test using GraphPad Prism software (v9).

## ChIP-sequencing

ChIP experiments were performed as above in duplicate, in accordance with ENCODE consortium guidelines (The_Encode_Consortium, 2017). Automated library preparation was performed at the Oxford Genomics Centre (University of Oxford) using the Apollo prep system (PrepX Comp ILMN 32i DNA Lib 96. cat no. 640102; Takara). Sequencing was performed on the NovaSeq 6000 platform,

using 150 bp paired-end configuration, according to Illumina protocols (Illumina).

### RNA-sequencing

RNA extraction was performed as above from six independent replicates, in accordance with ENCODE consortium guidelines (The_ENCODE_Consortium, 2016). PolyA+ RNA library preparation was conducted at GENEWIZ/Azenta Life Sciences using the NEBNext Ultra II Directional RNA Library Prep Kit for Illumina. Sequencing was performed on the NovaSeq platform, using 150 bp paired-end configuration, according to Illumina protocols (Illumina).

### Bioinformatic analysis of ChIP-seq data

#### *Preliminary analysis*
Illumina adaptor sequences were trimmed using TrimGalore (0.3.3). Reads were aligned to GRCh37 using BWA (0.7.5a-r405). Low-quality mapping was removed (MapQ < 15) using SAMtools (0.1.19) (56). Reads mapping to Duke Encode black list regions https://hgwdev.gi.ucsc.edu/cgi-bin/hgFileUi?db=hg19&g=wgEncodeMapability were excluded using BEDTools (2.17.0) (57). Duplicate reads were excluded using Picard tools (2.0.1) (http://broadinstitute.github.io/picard/). Read densities were normalised and expressed as reads per kilobase per million reads (RPKM) (58). One million random non-overlapping regions selected from ENCODE DNase Cluster II peaks (http://hgdownload.cse.ucsc.edu/goldenPath/hg19/encodeDCC/wgEncodeRegDnaseClustered/) were used as a control.

#### *Peak calling*
ChIP-seq peaks were identified using T-PIC (Tree shape Peak Identification for ChIP-Seq) (59) and MACS (model-based analysis of ChIP-seq) (60) in control mode. Peaks detected by both peak callers were filtered quantitatively using the total count under the peak to include only peaks that were above the 99.99th percentile of random background regions selected from the ENCODE DNase II cluster (*P*-value < 0.0001). Only peaks from each independent replicate that overlapped by at least 1 base pair (BEDTools 2.17.0 (57)) were considered.

#### *Quantitation of ChIP-seq signal*
Manipulation of .bam files was performed using SAMtools (0.1.19) (56). Briefly, SAMtools merge was used to merge sorted alignment .bam files from each replicate into a single .bam output file, which was then indexed using SAMtools index. SAMtools bedcov was then used to determine the read depth per bed region, which was normalised to the total reads in each dataset as determined using SAMtools flagstat.

#### *Normalisation of ChIP-seq signal*
ChIP-seq normalisation was performed using MAnorm2 (v1.0.0) (61). Briefly, genomic intervals were created using profile_bins from MAnorm2 with a bin size of 1,000 and shift size of 150 to count the reads that fall within each genomic interval. MA normalisation was then performed using "normalize" function from MAnorm2 with default options.

#### *Differential binding analysis*
Differential binding analysis for each comparison was performed using ChIPComp (v1.16.0) (62), and the Wald test *P*-value was corrected for multiple testing using the Benjamini–Hochberg procedure.

#### *Peak annotation*
The distance from each binding site to the nearest TSS in the Ensembl hg19 database was determined using PeakAnnotator (v1.4, https://www.bioinformatics.org/peakanalyzer/wiki/Main/Download) and plotted as a histogram in R (63).

#### *Data visualization*
Locus-specific data were visualized using Integrative Genomics Viewer (IGV; 2.14.13) (64). Chromosomal coordinates and gene annotation are from the RefSeq hg19 (GRCh37) build.

### Bioinformatic analysis of RNA-seq data

#### *Preliminary analysis*
Adaptor sequences were trimmed using TrimGalore (0.3.3). Reads were then aligned to GRCh37 using HISAT2 (2.0.5) (65), and non-uniquely mapped fragments were excluded using Picard tools (2.0.1) (http://broadinstitute.github.io/picard/). Total read counts for each UCSC-defined gene were extracted using HTSeq (0.5.4p3) (66) with "intersection-strict" mode, and significantly regulated genes were identified using DESeq2 (67).

#### *GSEA*
GSEA used 10,000 permutations, weighted ES, and pre-ranking of genes (68). Both differential expression significance according to DESeq2 and fold difference between the two conditions were used to rank genes according to the equation:

$$\pi_i = \varphi_i(-\log_{10} Pv_i),$$

where $\phi_i$ is the $\log_2$ fold change and $Pv_i$ is the *P*-value for gene *i* (69).

#### *TCGA RNA-seq data analysis*
To compare expression in ccRCC with normal kidney, gene expression levels were extracted from TCGA-KIRC level 3 RNA-seq data for 72 paired renal tumours and surrounding normal renal tissue and compared using a paired Wilcoxon rank sum test.

#### *Bioinformatic statistical analyses*
All statistical analyses were performed using R, version 4.0.5.

## Data Availability

High-throughput sequencing data are available at Gene Expression Omnibus, accession number GSE206029.

## Supplementary Information

# Acknowledgements

This study was supported by the National Institute for Health Research (DR Mole; NIHR-RP-2016-06-004), Cancer Research UK (DR Mole; A416016), the Deanship of Scientific Research, King Abdulaziz University, Ministry of High Education for Saudi Arabia (H Choudhry, DR Mole, and PJ Ratcliffe), the Ludwig Institute for Cancer Research (PJ Ratcliffe), the Wellcome Trust (PJ Ratcliffe; 106241/Z/14/Z), and the Francis Crick Institute (PJ Ratcliffe), which receives its core funding from Cancer Research UK (FC001501), the UK Medical Research Council (FC001501), and the Wellcome Trust (FC001501). The computational aspects of this research were supported by the Wellcome Trust Core Award Grant Number 203141/Z/16/Z and the NIHR Oxford BRC. The views expressed are those of the author(s) and not necessarily those of the NHS, the NIHR, or the Department of Health.

## Author Contributions

VN Lafleur: conceptualization, data curation, formal analysis, validation, investigation, visualization, methodology, and writing—original draft, review, and editing.
S Halim: data curation, software, formal analysis, validation, visualization, methodology, and writing—original draft, review, and editing.
H Choudhry: conceptualization, supervision, funding acquisition, and writing—review and editing.
PJ Ratcliffe: conceptualization, formal analysis, supervision, funding acquisition, investigation, and writing—review and editing.
DR Mole: conceptualization, resources, data curation, formal analysis, supervision, funding acquisition, validation, investigation, visualization, methodology, project administration, and writing—original draft, review, and editing.

## Conflict of Interest Statement

PJ Ratcliffe is a scientific co-founder of ReOx Ltd., a company, which is developing inhibitors of the HIF hydroxylase enzymes. The other authors declare no competing interest.

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
