## [Reviewer comments · Life Science Alliance]

Life Science Alliance

Multi-level interaction between HIF and AHR transcriptional pathways in kidney carcinoma

Veronique Lafleur, Silvia Halim, Hani Choudhry, Peter Ratcliffe, and David Mole

DOI: <https://doi.org/10.26508/lsa.202201756>

Corresponding author(s): David Mole, University of Oxford

Review Timeline:

Submission Date:	2022-10-06
Editorial Decision:	2022-10-31
Revision Received:	2022-12-23
Editorial Decision:	2023-01-09
Revision Received:	2023-01-11
Accepted:	2023-01-12

Scientific Editor: Novella Guidi

Transaction Report:

October 31, 2022

Re: Life Science Alliance manuscript #LSA-2022-01756-T

Prof. David Robert Mole
University of Oxford
The NDM Research Building
Roosevelt Drive
Headington
Oxford OX3 7FZ
United Kingdom

Dear Dr. Mole,

Thank you for submitting your manuscript entitled "Multi-level interaction between HIF and AHR transcriptional pathways in kidney carcinoma" to Life Science Alliance. The manuscript was assessed by expert reviewers, whose comments are appended to this letter. We invite you to submit a revised manuscript addressing the Reviewer comments.

Thank you for this interesting contribution to Life Science Alliance. We are looking forward to receiving your revised manuscript.

Sincerely,

B. MANUSCRIPT ORGANIZATION AND FORMATTING:

Reviewer #1 (Comments to the Authors (Required)):

The paper by Lafleur et al. addresses an important and timely topic, namely the interaction between HIF and AHR signaling pathways. Both transcription factors HIF and AHR are closely linked due to the same dimerization partner HIF-1 β /ARNT. Thus, it has been a question for many years whether within these two signaling pathways HIF and AHR compete for HIF-1 β /ARNT, and thus influence activation of each other.

The authors now report uni- and bidirectional competition depending on cellular context, in this case cell lines, as well as cooperative effects on the activity of both transcriptional complexes depending on the proximity of the respective response elements in the DNA. Finally, the authors address the question whether the antagonism between HIF and AHR may be relevant in clear cell renal carcinoma where recently inhibitors for HIF-2 have been introduced into therapy.

In general, this is a well-written manuscript and the data support the interpretation put forward by the authors.

My questions are as follows:

1. Fig. 2 A

I assume *ns stands for non-specific in the HIF-1 β blot. Do the authors have any idea about this unspecific band? Which antibody did the authors use? Usually HIF-1 β gives a clear signal, which should be visible in the input fraction and then obviously higher in the IP. Please comment.

2. In HepG2 cells the authors found unidirectional inhibition of AHR target gene expression by hypoxia but not vice versa. Have the authors checked for highly sensitive target genes like erythropoietin? Potentially, the inhibition may be different for finally tuned HIF target genes versus other HIF targets.

3. When analyzing RNA sequences from ccRCC tumors the authors excluded genes that were upregulated by HIF and AHR. Why? With respect to the potential relevance in kidney cancer, wouldn't it be most interesting to look specifically into those genes? Can the authors provide data such as in Fig. 6 for genes with overlap between the two response elements?

Reviewer #2 (Comments to the Authors (Required)):

In the current paper the authors dealt with the question whether hypoxia-inducible factor (HIF) and aryl hydrocarbon receptor (AHR) that share a common HIF-1 β /ARNT subunit and similar DNA-binding motifs, are involved in a crosstalk between the hypoxia response and xenobiotic response transcriptional pathways. The authors found that both general and locus-specific mechanisms of interaction between HIF and AHR exist which allow antagonistic as well as cooperative actions. This is an interesting study and important for the field. Overall, the scientific approaches and data presentation is convincing and includes appropriate controls. The replication of all experiments in each respective cell line provides a thorough palette for comparison.

My only suggestion would be to include size markers.

Reviewer #1 (Comments to the Authors (Required)):

The paper by Lafleur et al. addresses an important and timely topic, namely the interaction between HIF and AHR signaling pathways. Both transcription factors HIF and AHR are closely linked due to the same dimerization partner HIF-1 β /ARNT. Thus, it has been a question for many years whether within these two signaling pathways HIF and AHR compete for HIF-1 β /ARNT, and thus influence activation of each other.

The authors now report uni- and bidirectional competition depending on cellular context, in this case cell lines, as well as cooperative effects on the activity of both transcriptional complexes depending on the proximity of the respective response elements in the DNA. Finally, the authors address the question whether the antagonism between HIF and AHR may be relevant in clear cell renal carcinoma where recently inhibitors for HIF-2 have been introduced into therapy.

In general, this is a well-written manuscript and the data support the interpretation put forward by the authors.

My questions are as follows:

1. Fig. 2 A

I assume *ns stands for non-specific in the HIF-1β blot. Do the authors have any idea about this unspecific band? Which antibody did the authors use? Usually HIF-1β gives a clear signal, which should be visible in the input fraction and then obviously higher in the IP. Please comment.

The reviewer is correct that *ns designates a non-specific band seen on the HIF-1b immunoblots in Figure 2 and Supplemental Figure 3, which used the Novus Biologicals NB-100-124 mouse monoclonal antibody. This antibody was used as it was from a different species to the immunoprecipitating antibody (Novus Biologicals rabbit pAb, NEB-100-110) and so the (anti-mouse immunoglobulin) secondary antibody would not cross-react with the immunoprecipitating antibody. In independent experiments using a HIF-1β CRISPR knockout clone (in SH-S5Y5 cells), we have confirmed that HIF-1β corresponds to the upper band migrating at 100 kDa (see below). Furthermore, the HIF-1b immunoprecipitating antibody (Novus Biologicals rabbit pAb, NEB-100-110) only recognised the assigned upper band. In supplemental Figures 1 and 2, in which immunoprecipitations were not performed, we used the Cell Signalling rabbit pAb, #5537, which gave a clear single band. We have amended the manuscript to make this clearer.

2. In HepG2 cells the authors found unidirectional inhibition of AHR target gene expression by hypoxia but not vice versa. Have the authors checked for highly sensitive target genes like erythropoietin? Potentially, the inhibition may be different for finely tuned HIF target genes versus other HIF targets.

We examined the expression of NDRG1 as an example of a highly sensitive HIF-target gene since it is expressed in all the cell-lines studied, whereas erythropoietin is only expressed in HepG2 cells. However, following the reviewer's suggestion, we have examined the expression of erythropoietin (EPO) in HepG2 cells (see below). Similar to NDRG1, we did not observe inhibition of erythropoietin expression by ITE-treatment in HepG2 cells. Furthermore, the expression of two additional sensitive HIF-target genes, PFKFB4 and PLOD2 was not inhibited by ITE in HepG2, HKC8 or PC3 cells, but was inhibited in the renal cancer cell lines.

3. When analyzing RNA sequences from ccRCC tumors the authors excluded genes that were upregulated by HIF and AHR. Why? With respect to the potential relevance in kidney cancer, wouldn't it be most interesting to look specifically into those genes? Can the authors provide data such as in Fig. 6 for genes with overlap between the two response elements?

Genes that were upregulated by both HIF and AHR were excluded from this analysis as an increase in HIF activity and consequent fall in AHR activity at these gene loci would be predicted to have opposing effects on their expression. However, we agree that with respect to kidney cancer, it may be important to determine whether one effect dominates over the other at these common loci. We therefore examined the expression of common HIF and AHR target genes, individually and using a composite score. In contrast to genes regulated by the individual factors, expression of these genes did not differ significantly between clear cell kidney cancer and normal kidney, either individually (not shown) or based on the composite score. Furthermore, no significant association was seen between the composite score and VHL copy number alteration in the tumours. Taken together, this suggests that the increase in HIF activity at these loci in kidney cancer is balanced by the reduction in AHR activity. We have included these new analyses in a revised Figure 6.

Reviewer #2 (Comments to the Authors (Required)):

In the current paper the authors dealt with the question whether hypoxia-inducible factor (HIF) and aryl hydrocarbon receptor (AHR) that share a common HIF-1 β /ARNT subunit and similar DNA-binding motifs, are involved in a crosstalk between the hypoxia response and xenobiotic response transcriptional pathways. The authors found that both general and locus-specific mechanisms of interaction between HIF and AHR exist which allow antagonistic as well as cooperative actions

This is an interesting study and important for the field. Overall, the scientific approaches and data presentation is convincing and includes appropriate controls. The replication of all experiments in each respective cell line provides a thorough palette for comparison. My only suggestion would be to include size markers.

We have now added size markers to the immunoblots

January 9, 2023

RE: Life Science Alliance Manuscript #LSA-2022-01756-TR

Prof. David Robert Mole
University of Oxford
The NDM Research Building
Roosevelt Drive
Headington
Oxford OX3 7FZ
United Kingdom

Dear Dr. Mole,

Thank you for submitting your revised manuscript entitled "Multi-level interaction between HIF and AHR transcriptional pathways in kidney carcinoma". We would be happy to publish your paper in Life Science Alliance pending final revisions necessary to meet our formatting guidelines.

- please add the Twitter handle of your host institute/organization as well as your own or/and one of the authors in our system
- please use the [10 author names, et al.] format in your references (i.e. limit the author names to the first 10)
- please add your supplemental figure legends to the main manuscript text
- please add a figure callout for Figure 2B and Figure S2B-D

A. FINAL FILES:

B. MANUSCRIPT ORGANIZATION AND FORMATTING:

Sincerely,

Reviewer #1 (Comments to the Authors (Required)):

The authors have addressed my questions to my full satisfaction. I have no further comments.

January 12, 2023

RE: Life Science Alliance Manuscript #LSA-2022-01756-TRR

Prof. David Robert Mole
University of Oxford
The NDM Research Building
Roosevelt Drive
Headington
Oxford OX3 7FZ
United Kingdom

Dear Dr. Mole,

Thank you for submitting your Research Article entitled "Multi-level interaction between HIF and AHR transcriptional pathways in kidney carcinoma". It is a pleasure to let you know that your manuscript is now accepted for publication in Life Science Alliance. Congratulations on this interesting work.

DISTRIBUTION OF MATERIALS:

Again, congratulations on a very nice paper. I hope you found the review process to be constructive and are pleased with how the manuscript was handled editorially. We look forward to future exciting submissions from your lab.

Sincerely,
